# Urban Growth and Habitat Connectivity: A Study on European Countries

Francesco Zullo *, Cristina Montaldi, Gianni Di Pietro and Bernardino Romano

Department of Civil, Construction-Architectural and Environmental Engineering, University of L'Aquila, Piazzale Ernesto Pontieri 1, Monteluco di Roio, 67100 L'Aquila, Italy
* Correspondence: francesco.zullo@univaq.it; Tel.: +39-0862-434104

**Abstract:** The main tool for biodiversity conservation at the European level is the Natura 2000 network. The identification of Natura 2000 as an "ecological network spread over the entire European Union territory" is the symbolic image launched by the Habitat Directive (92/43/EEC) even though many considerations focused on the contradiction between the shared model of the ecological network—based on spatial continuity—and the fragmented geographical configuration of the Natura 2000 sites. Currently, it stretches across all 28 European countries, both on land and at sea, and it is made up of over 27,000 sites for a total extension of approximately 1,150,000 km². The land area covered by N2000 corresponds to approximately 18% of the total EU, with the national coverage ratio ranging from a minimum of 9% to a maximum of 38% in the various European countries. The aim of this study was to determine the degree of landscape fragmentation caused by the urban areas towards the Natura 2000 network, with the aim of analyzing how the current urban settlements' geography could compromise their functionality. The proximity analysis carried out provides the necessary information to achieve full efficiency in the connections between the different habitats. In addition, these results give indications on which planning scale is most appropriate to intervene to reduce environmental fragmentation.

**Keywords:** proximity analysis; Natura 2000 network; urbanization; landscape fragmentation; indicators engineering

## 1. Introduction

The environmental network concept constitutes a paradigm shift in nature conservation policies, which have effectively progressed from safeguarding individual protected areas to conserving whole national ecosystem structures [1–4]. At a European level, this has been consolidated over time thanks in part to important European community-level initiatives such as the EECONET (The European Ecological Network) program or PEBLDS, the Pan European Biological and Landscape Diversity Strategy, whose purpose is to implement a Europe-wide Convention of Biological Diversity (CBD). In both cases the key element is the development of the Pan European Ecological Network (PEEN), the goal of which is to conserve eco-systems, habitats, and species of importance in Europe. PEEN has taken the form of three projects covering central and eastern, south-western, and western Europe, respectively.

Set up in accordance with the Habitat Directive (92/43/EEC) for the purposes of ensuring the long-term conservation and maintenance of natural habitats and flora and fauna species, the Natura 2000 network continues in this direction and is the key European Union biodiversity conservation policy tool. The N2000 network constitutes a disseminated ecology network covering over 27,000 sites, which, overall, cover over 1 million square kilometers, almost one-fifth of the European land area and a significant part of the seas around it. It is one of the largest coordinated conservation zone networks in the world. The national coverage rate in the various European states varies from a minimum of 9% to a



maximum of 38%. This difference derives from both the Mediterranean region's greater habitat and species diversity [5], as compared to the Atlantic, and higher intensive land use densities and environmental fragmentation in certain countries [6–8]. It should also be highlighted that the Natura 2000 network is a dynamic one and that the site number and extension thus varies. To date, as the European Commission has underlined, the environmental network is virtually complete on land whilst the marine environment is currently the subject of significant number of works, which is complicated by the limited availability of scientific information on protected marine species and habitat distribution at a detailed enough level to allow for sites and management systems to be identified.

Large parts of Europe have become fragmented because of the expansion of urban and transport infrastructure, with important effects on the habitat connectivity. As underlined by the European Environment Agency (https://www.eea.europa.eu/ims/landscape-fragmentation-pressure-in-europe (accessed on 8 August 2022)), every $km^2$ in the 27 EU Member States (EU-27+UK) comprises around 1.4 habitats and 27% of the land is considered highly fragmented.

In the European context, it should be noted that research into the ecology networks has, in any case, activated a spectrum of knowledge that has fed into other sectors, including the greenways and ecosystem services cited above [8,9]. Knowledge of anthropogenic pressure together with existing levels of environmental fragmentation is now one of the indispensable pre-requisites both for the functioning effectiveness of the network and to guarantee its long-term conservation [10–15]. The proximity analysis used in this work is a useful tool for the purposes of a first careful analysis of the degree of anthropogenic impairment of the natural matrix at various distances from the sites making up the network's structure.

## 2. Study Area

This study analyzed the level of environmental fragmentation caused by urban areas in the Natura 2000 network structure in European countries (Figure 1) through indicators engineering techniques. The European Natura 2000 network includes over 27,000 sites that cover a 1,150,000 $km^2$ surface, of which about 935,000 $km^2$ are terrestrial.

Table 1 shows the main statistical parameters relating to the size of the Natura 2000 network in the analyzed countries. Although the network includes both terrestrial and marine sites, only continental ones were considered for this study. In numerical terms, Germany alone hosts about 20% of the total number of sites, the average extension of which is 14 $km^2$, reaching 20% of the German territory extent. The lower coverage value is recorded in the United Kingdom (1.1%), where there are less than 800 sites, many of which are small (first quartile of 8.5 ha). Coverage lower than 20% is detected in Scandinavian countries but also in France and The Netherlands. Specifically, the latter has a very limited number of sites compared to other Western European countries that are distributed over highly populated territories. The Iberian Peninsula is the area with the largest extension of the network: just under 200,000 $km^2$ with a national coverage that reaches 35% of the territory for Spain and 25% for Portugal. The latter is the country with the lowest number of sites (94), but the average extension is among the highest in Europe (about 230 $km^2$) and the variability of its surface is small (coefficient of variation just over 1). Eastern countries have a very high value of coverage, except for Baltic counties (Latvia, Estonia, and Lithuania), the coverage of which is lower than 20%. The remaining countries have significantly higher values: Hungary and Poland are close to 30% while countries such as Bulgaria and Croatia have over half of their territory covered by N2000 sites.

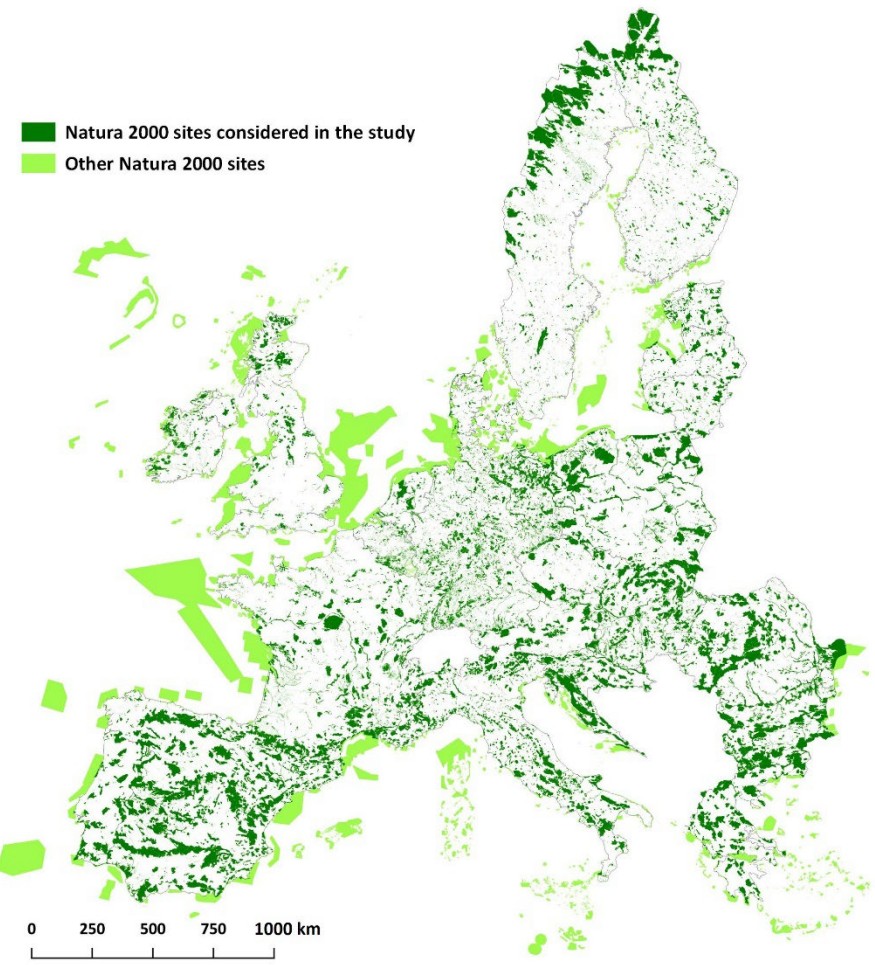

**Figure 1.** European N2000 network geography.

**Table 1.** The main statistical parameters of the Natura 2000 sites in the analyzed European countries. The background color shows the cluster of European countries based on N2000 cover Ratio (Ascending order).

| | | | | Natura 2000 Network | | | | | | |
|---|---|---|---|---|---|---|---|---|---|---|
| Country | Continental Sites | Continental Sites Area (kmq) | Continental Country Area (kmq) | N2K Cover Ratio (%) | Mean (ha) | Standard Deviation | Coefficient of Variation | Median (ha) | First Quartile (ha) | Third Quartile (ha) |
| United Kingdom | 762 | 2611.44 | 232,855.87 | 1.12 | 342.7 | 1390.15 | 4.06 | 31.94 | 8.426 | 138.27 |
| Latvia | 332 | 7447.18 | 64,586.73 | 11.53 | 2243.13 | 8033.8 | 3.58 | 263.15 | 67.67 | 1027.89 |
| Sweden | 3641 | 55,157.58 | 44,2296.83 | 12.47 | 1514.9 | 14,869.8 | 9.82 | 31.22 | 8.14 | 130.41 |
| Finland | 1702 | 42,499.92 | 332,956.68 | 12.76 | 2497.05 | 17,729.06 | 7.10 | 149.81 | 37.001 | 605.32 |
| Ireland | 536 | 11,315.5 | 69,473.92 | 16.29 | 2111.1 | 6800.3 | 3.22 | 249.47 | 64.01 | 904.46 |
| France | 1630 | 89,870.18 | 539,648.6 | 16.65 | 5513.5 | 13,430.35 | 2.43 | 1371.97 | 282.28 | 5371.8 |
| Lithuania | 550 | 10,846.37 | 64,800.81 | 16.74 | 1972.06 | 6552.89 | 3.32 | 177.02 | 55.51 | 692.865 |
| Austria | 352 | 15,274.61 | 83,939.91 | 18.20 | 4339.38 | 13,341.19 | 3.07 | 117.66 | 23.54 | 1331.02 |
| Belgium | 310 | 5705.51 | 30,663.73 | 18.61 | 1840.49 | 6498.65 | 3.53 | 689 | 292.94 | 1913.76 |
| Czech Republic | 1153 | 14,956.63 | 78,873.25 | 18.96 | 1297.19 | 8253.55 | 6.36 | 26.13 | 4.23 | 158.66 |
| The Netherlands | 180 | 7065.71 | 36,236.38 | 19.50 | 3925.39 | 13,442.05 | 3.42 | 965 | 304.44 | 2477.82 |
| Germany | 5155 | 71,463.51 | 355,745.52 | 20.09 | 1386.29 | 4278.53 | 3.09 | 224.36 | 54.55 | 893.941 |
| Estonia | 444 | 9202.12 | 41,189.82 | 22.34 | 2072.55 | 5906.41 | 2.85 | 166.64 | 32.3 | 820.48 |
| Italy | 2175 | 58,778.7 | 250,393.97 | 23.47 | 2702.47 | 8400.04 | 3.11 | 493.13 | 118.25 | 1814.39 |
| Portugal | 94 | 21879.07 | 88,735.27 | 24.66 | 23,275.61 | 28,680.21 | 1.23 | 11,796.12 | 2106.74 | 33,222.45 |
| Poland | 984 | 79,292.9 | 311,877.88 | 25.42 | 8058.22 | 23,713.43 | 2.94 | 854.136 | 129.58 | 4766.84 |
| Hungary | 525 | 26,525.84 | 93,009 | 28.52 | 5052.54 | 12,568.59 | 2.49 | 852.82 | 252.9 | 3635.92 |
| Romania | 604 | 77,565.56 | 238,362.38 | 32.54 | 12,841.98 | 35,602.47 | 2.77 | 2131.95 | 359.77 | 10,865.52 |
| Greece | 259 | 38,156.32 | 110,093.82 | 34.66 | 14,732.17 | 19,841.97 | 1.35 | 7763.39 | 1710.86 | 20,238.65 |
| Spain | 1417 | 172,824.77 | 493,461.32 | 35.02 | 12,196.52 | 26.213 | 2.15 | 2383.22 | 349.667 | 11,658.22 |
| Slovakia | 683 | 19,222.49 | 49,023.68 | 39.21 | 2814.42 | 11,187.98 | 3.98 | 83.38 | 20.39 | 354.47 |
| Bulgaria | 338 | 56,033.66 | 110,994.23 | 50.48 | 16.578 | 33,696.58 | 2.03 | 3548.15 | 882 | 15,446.37 |
| Croatia | 486 | 29,111.95 | 52,427.74 | 55.53 | 5990.11 | 22,307.77 | 3.72 | 87.776 | 2.469 | 1145.126 |
| Slovenia | 355 | 11,682.67 | 20,271.57 | 57.63 | 3290.89 | 12,221.09 | 3.71 | 89.62 | 13.19 | 792.02 |

## 3. Materials and Methods

The data used in this work came from a range of sources. The land cover maps came from the Copernicus portal-Land Monitoring Service (https://land.copernicus.eu/pan-european/corine-land-cover (accessed on 8 August 2022)). The minimum mapping unit (MMU) is 25 hectares for areal phenomena with a minimum width of 100 m for linear ones. These data are available for a 30-year time frame (from 1990 to 2018). The method of acquisition was the same over the analyzed period and the data are thus fully comparable. This information base was used to extract urbanized areas of 2000 and 2018. Urbanized areas are those whose land use is urban and whose natural coverage has been replaced or maintained, built up land, and complementary areas, such as public and private gardens, sports facilities, gravel roads, and other service areas. Non-urban road systems fall outside these calculations. These surfaces are those covered by item 1 of the CORINE Land Cover first level classification. The data relating to the geography of Natura 2000 network sites comes from the website of the European Environment Agency (https://www.eea.europa.eu/data-and-maps/data/natura-13 (accessed on 8 August 2022)). The network consists of SPAs (Special Protection Areas or SPAs defined by the Birds Directive), SCIs and SACs (Sites of Community Importance and Special Areas of Conservation or SACs defined by the Habitats Directive, respectively). This study solely considered sites covering land areas larger than 2 hectares. In the database, to each site, a code indicating the country of belonging is assigned, allowing identification of the Natura 2000 sites' geography. For each country, until a 5 km distance, a proximity analysis was carried out through the definition of 5 buffers 1 km from each other. For each of them, through the urbanization density and the urban variation rate, the degree of anthropogenic interference was analyzed. The purpose of using these indicators is twofold: on the one hand, it allows the identification of models that describe the degree of insularization of the sites while, on the other hand, through the comparison with the values of the FRD (Fragmentation Reduction Distance) index, it allows the effectiveness of possible measures to reduce environmental fragmentation at different scales to be established. The FRD index has already been used in several local and national studies [10–12]. In this work, it was used for the first time on the European scale and compared directly with existing levels of urban fragmentation.

As mentioned above, this provides an updated picture of the effectiveness of the necessary fragmentation reduction measures. The formulations of the used indexes are as follows:

$$\textit{Urban Density } \mathrm{UD} = \frac{\sum \mathrm{Aurb}}{\mathrm{Au}} \ (\%) \tag{1}$$

$$\textit{Fragmentation Reduction Rate } \mathrm{FRR} = \frac{\mathrm{Np(1)}}{\mathrm{Np(1+i)}} \ (\%) \tag{2}$$

$$\textit{Urban Variation Ratio } \mathrm{UVR} = \frac{\mathrm{Aurb2018 - Aurb2000}}{\mathrm{Aurb2000}} \ (\%) \tag{3}$$

where

Aurb = urbanized area;
Au = considered territorial entity;
Np (1) = number of patches deriving from the aggregation with order 1 buffer;
Np (1 + i) = number of patches deriving from the aggregation with order 1 + i buffer.

The FRR index expresses the degree of reduction in environmental fragmentation. The FRR value progressively decreases with the increase in the buffer distance. Starting from the initial conditions of fragmentation in each country, defined by the number of N2K sites, external buffers were drawn at fixed and increasing distances (1 km). Each time a buffer was drawn around all patches, their number was reduced. This made it possible to correlate the buffer distances and the number of matching patches until the extreme value of one patch was reached when all the original patches were merged. Moreover, from this indicator, it was possible to derive, in an indirect way, the FRD50 index. The latter

indicates the distance at which spatial reconnection interventions can be operated to reduce the existing environmental fragmentation by 50%. In essence, this value indicates the level of spatial detail necessary for improving the environmental continuity between the sites of the network, as shown in Figure 2.

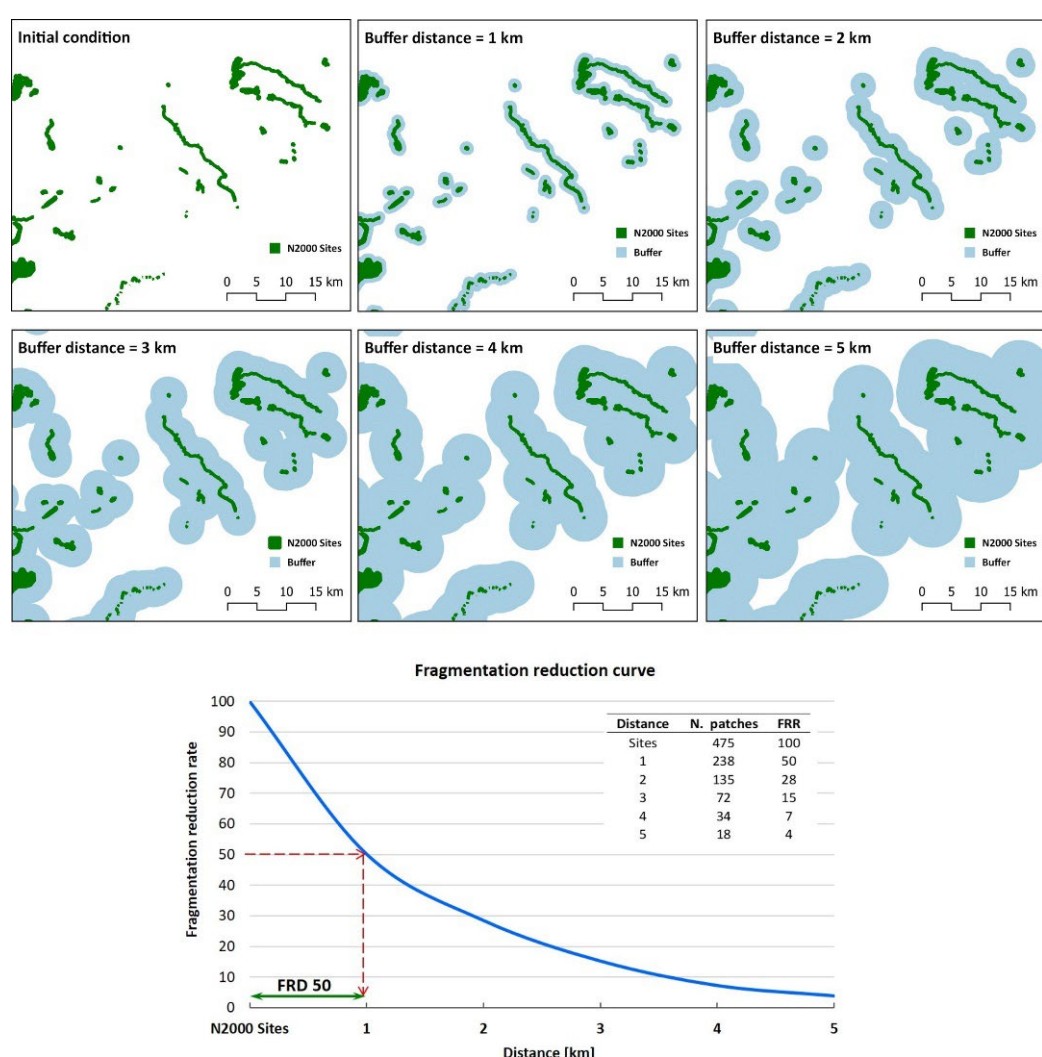

**Figure 2.** Example of the calculation method of the FRR and FRD indicator.

## 4. Results

The curves in Figures 3 and 4 show the Natura 2000 network and buffer distances (from 1 to 5 km) on the x-axis and the values of the UD index on the left (2018) and those of the UVR on the right on the double y-axis. The dotted line shows the national UD values for 2018.

The two curves' trend shows the anthropogenic pressure on the Natura 2000 network in the European countries. In general, there is a very low UD value within the network, with percentages that are often less than 1% and that, only in some cases, are exceeded (Croatia and Poland in the east and Austria and Belgium in the west, with the latter reaching 5%). A common condition in all the countries is the high value of the UD index in the 1 km buffer adjacent to the Natura 2000 sites. In this belt, the highest values are reached, and these values are often double those inside the network. The Netherlands and Belgium achieve a large value of almost 20%. In the western countries, UD is mostly equal to 10% in addition to the eastern counties such as Bulgaria, Romania, Hungary, the Czech Republic, and Slovakia. A different condition is detected in Sweden and Finland, where the UD index value is extremely modest throughout the study area. In this case, the climatic

and morphological conditions play an important role in the definition of the settled areas' geography. Joint analysis of the curves shown in Figures 3 and 4 show five prevailing insularity conditions. The identified types are summarized in Figure 5.

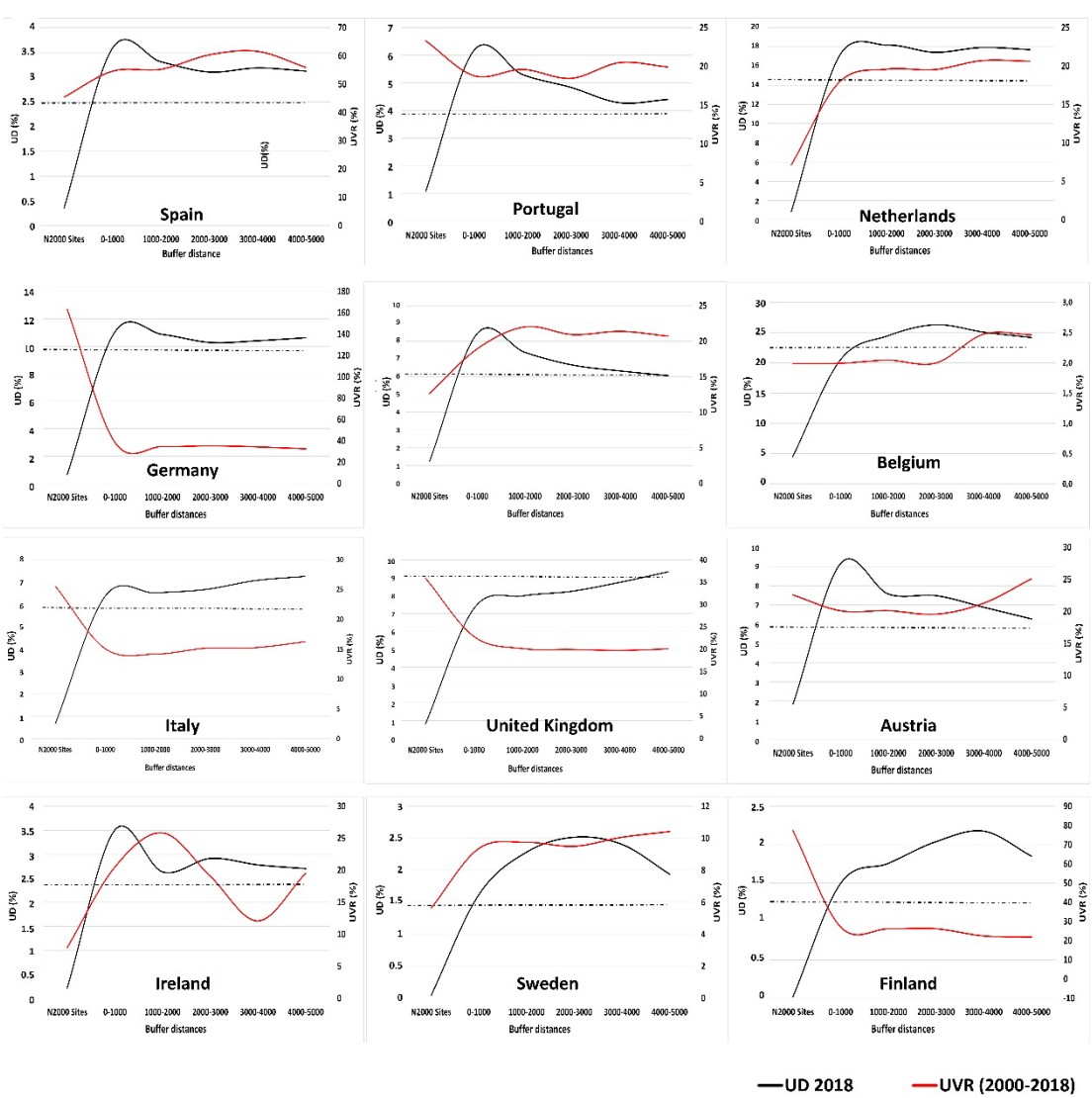

**Figure 3.** Values of the UD and UVR indices in Western European countries. The x-axis shows the analyzed buffer distances (N2000 sites, 1, 2, 3, 4, and 5 km) while the y-axis shows the relative UD (on the left y-axis) and UVR (on the right y-axis) values.

The first, named "open", is typical of Scandinavian countries, for whom the level of anthropogenic pressure determined by biotic flows is not relevant in terms of both close proximity and at long distances from the sites, and the national urban density is lower than the European one (5%). Ireland, Spain, Greece, and the Baltic countries belong to the "semi-open" group. In this case, there is a limited degree of anthropogenic pressure near the sites (UD < 4–5%), followed by lower values at long distances. In other countries, the values measured close to the sites are under conditions of a strong urban siege on the N2000 network sites. To identify possible different conditions, they were described with the help of three sub-models. In the first group, named "gated", a medium to high level of anthropogenic pressure is detected close to the site and the maximum UD is in this area, with a value ranging from 5% and 10%. In the following buffers (>2 km), this value decreases significantly. Countries in this group include Austria, France, Romania, and Portugal, for which the UD values at long distances (5 km) tend asymptotically to

the national average. The model called "trapped" reports the condition in which the sites are substantially immersed in the surrounding urban matrix, with UD values between 8% and 10%, both in proximity and at greater distances. Countries such as Germany, Poland, Hungary, and the Czech Republic are included in this model. The model defined as "packed" differs from the previous one since the already high UD index value close to the sites progressively increases over long distances (Italy and United Kingdom) or remains stable around extremely high values (Belgium and The Netherlands), with a progressive compromise in the territorial matrix surrounding the protected areas.

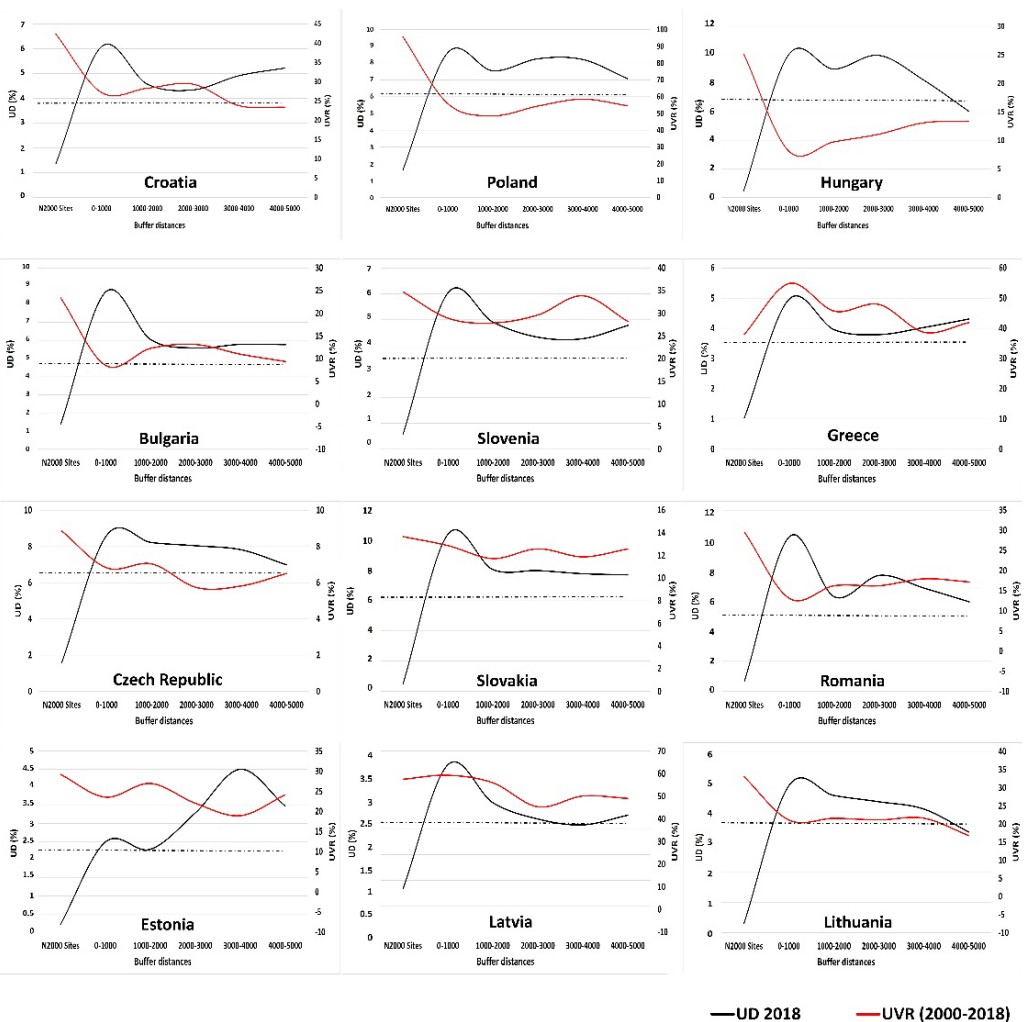

**Figure 4.** Values of the UD and UVR indices in Eastern European countries. The x-axis shows the analyzed buffer distances (N2000 sites, 1, 2, 3, 4, and 5 km) while the y-axis shows the relative UD (on the left y-axis) and UVR (on the right y-axis) values.

The analysis of the CLC data shows that between 2000 and 2018, the growth of urbanized areas increased to over 1600 km$^2$, which is the same as a square with a 40 km side. Additionally, in this case, clear differences are observed among the different countries. About half of the urban transformation inside the N2000 sites involved the territory of the German-Polish plain, where over 700 km$^2$ was converted to urban use at a speed of 43 km$^2$ per year, equivalent to about 12 ha/day. In other countries, the transformative energy is still significant but weaker. For example, Spain and Bulgaria transform 150 and 100 km$^2$ of land inside the protected areas, respectively. On the other hand, there are cases in which soil has been safeguarded since the recorded urban changes are less than 10 km$^2$. This includes the Scandinavian states and The Netherlands and Belgium while in Eastern Europe, the same condition was found for Estonia, Slovakia, and Lithuania.

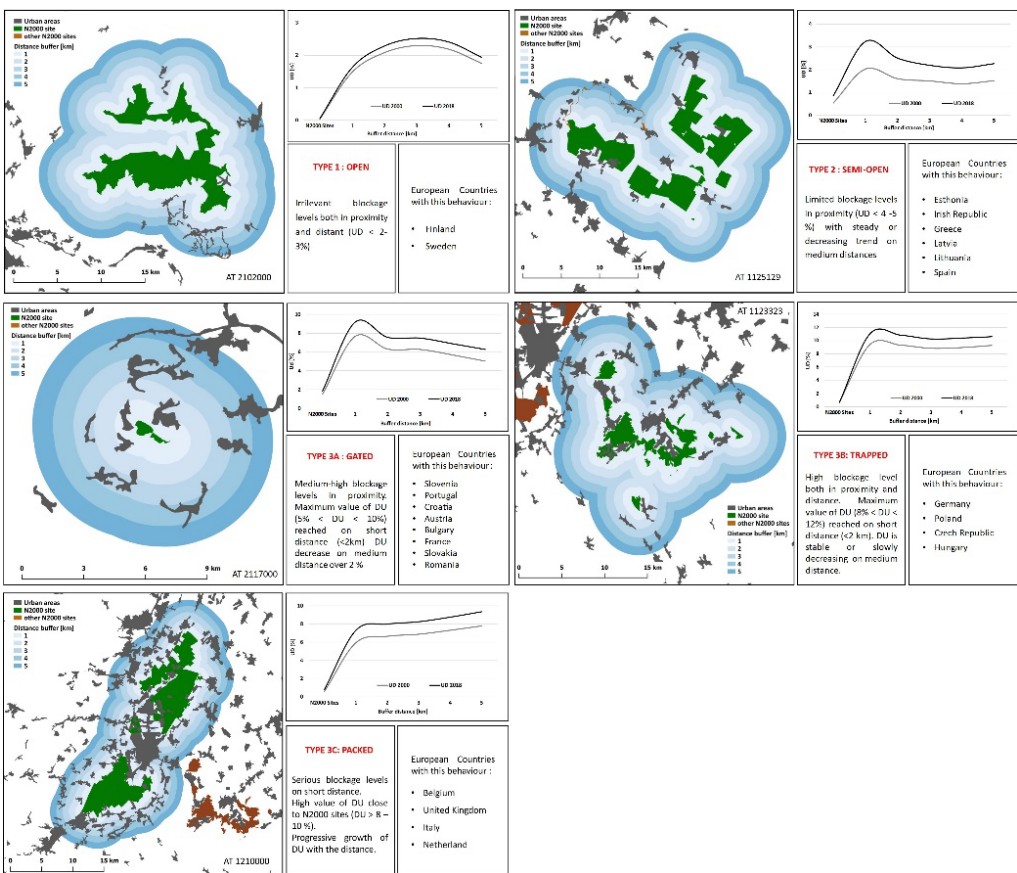

**Figure 5.** Insularity models of the European countries.

Further analysis, based on the use of the FFR index, made it possible to evaluate the relationship between the reciprocal distances of the network sites and the reduction in the current level of fragmentation, thus identifying the scale of intervention. According to the calculation methodology expressed in the Materials and Methods section, the trend of the fragmentation reduction curves (Figures 6 and 7) allows identification of the spatial organization of the N2000 sites in the territory of the country under investigation. For some countries (Belgium, Croatia, Germany, Hungary, Ireland, and Slovenia), work over short distances to connect Natura 2000 sites is very low, which are already in a pseudo-aggregate form. In the opposite situation, for the Scandinavian countries, Latvia, Lithuania, and Portugal, it is necessary to work with spatial reconnection interventions at wider territorial scales since the distances between the sites increase and the environmental matrix is largely disconnected. The other countries show intermediate conditions compared to those described. In the territories of Austria, the Czech Republic, Italy, Poland, and Romania, the N2000 sites are grouped close together (poorly transformed environmental matrix), with the remaining being the most isolated sites. This is evidenced by the fact that the curves reach low FRR values at high distances of higher than one kilometer. Similar conditions exist in the remaining countries (Bulgaria, Estonia, France, Greece, The Netherlands, Slovakia, Spain, and the United Kingdom), in which, unlike before, the curve decreases quickly and the FRR index reaches low values over medium to short distances. Table 2 resumes the described countries conditions and it shows the example models of relative N2K pattern.

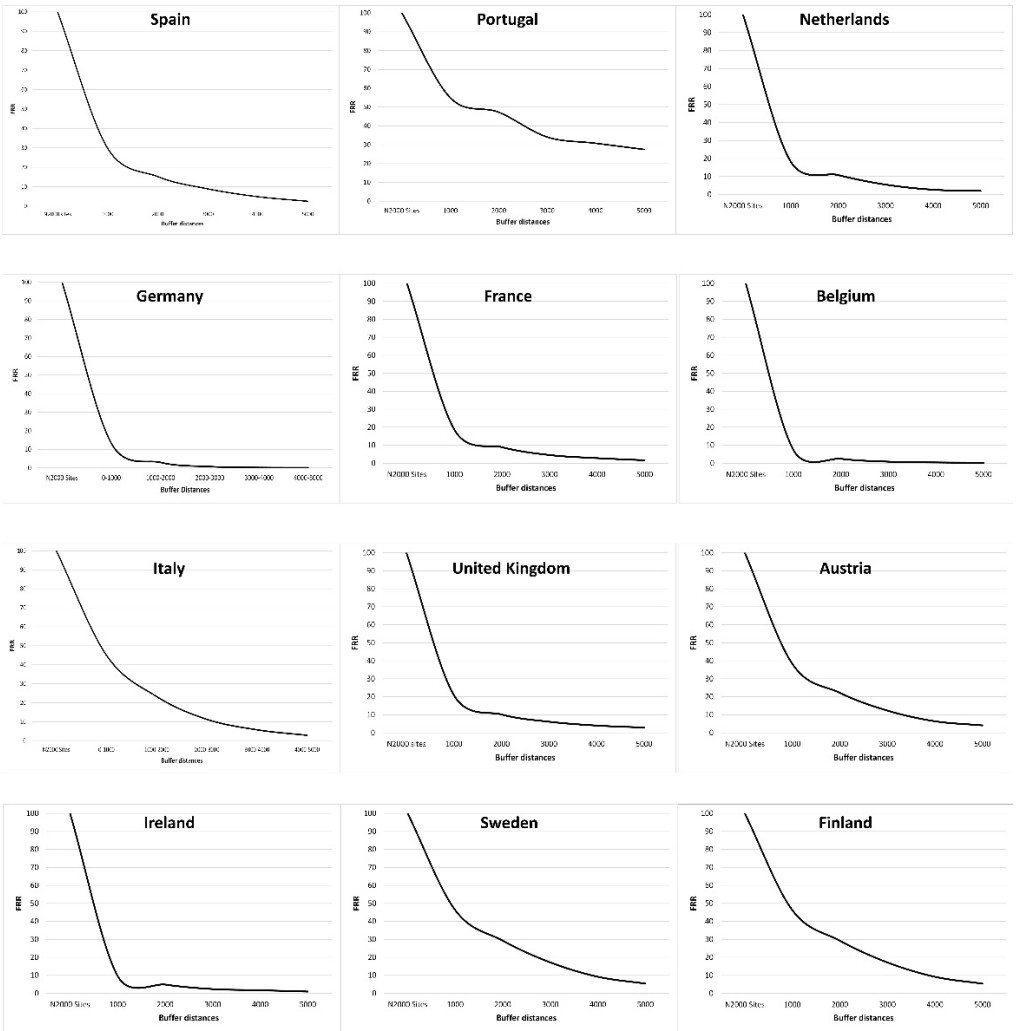

**Figure 6.** Fragmentation curves for Western European countries. The x-axis shows the analyzed buffer distances (N2000 sites, 1, 2, 3, 4, and 5 km) while the y-axis shows the relative FRR values.

As indicated above, the analysis of the FRR curve made it possible to calculate the FRD index, which allowed the possible scale of environmental defragmentation interventions to be set. The chart in Figure 8 shows the FRD50 values together with the UD index evaluated in the two belts 1 km from the protected sites. In particular, the FRD50 values of a few tens of meters and up to 150–200 m indicate the possibility of opting for defragmentation interventions on an urban scale, with very limited and localized actions. For higher FRD50 values, it is necessary to act at the planning level and therefore on a larger scale, with much more complex problems that can include, for example, the re-motion and lightening of barriers, especially for FRD50 over one kilometer. As is evident in the chart, the most critical situation appears to be that of Belgium, which has an extremely low FRD50 (<20 m) but, at the same time, the highest UD values. A similar condition was found in the United Kingdom, where the FRD50 index value is higher (just under 400 m) but the anthropic pressure near the protected sites is very high. The fact that the FRD50 value is low is not sufficient on its own to ensure the possibility of reconnecting ecosystems. The strongly transformed spatial matrix in which the sites are immersed in creates many problems in the identification of free and suitable corridors for connecting them.

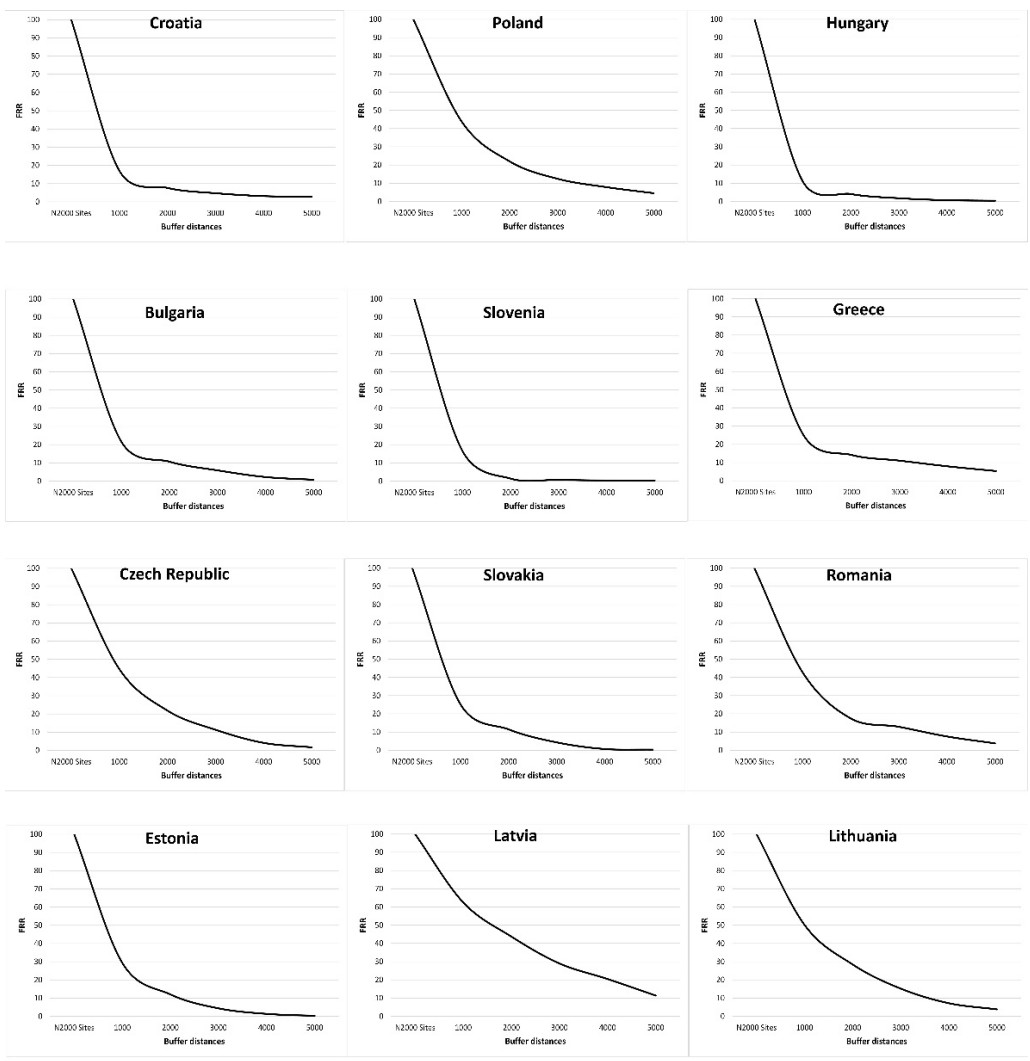

**Figure 7.** Fragmentation curves for Eastern European countries. The x-axis shows the analyzed buffer distances (N2000 sites, 1, 2, 3, 4, and 5 km) while the y-axis shows the relative FRR values.

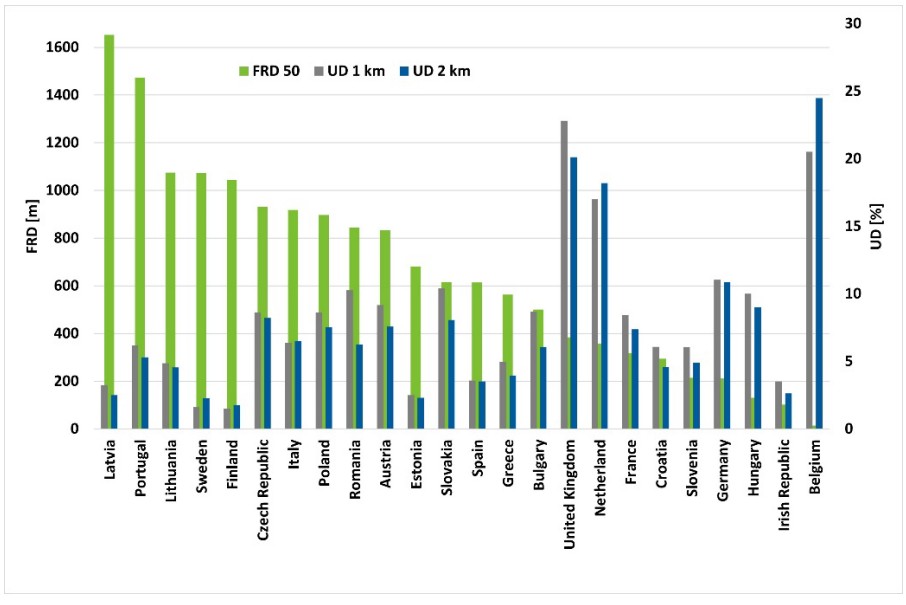

**Figure 8.** FRD50 and UD in the first 2 km from the Natura 2000 sites.

**Table 2.** Sampling of the fragmentation reduction curves.

| Model | Typical FRR Curve | Pattern | Countries |
|:-----:|:-----------------:|:-------:|:---------:|
| A |  |  | Belgium, Croatia, Germany, Hungary, Ireland, and Slovenia |
| B |  |  | Bulgaria, Estonia, France, Greece, The Netherlands, Slovakia, Spain, and the United Kingdom |
| C |  |  | Austria, Czech Republic, Italy, Poland, and Romania |
| D |  |  | Finland, Latvia, Lithuania, Portugal, and Sweden |

Then, there are countries such as Ireland, France, and Slovenia, where spatial reconnection at the urban scale could increase the environmental continuity between the various sites since the level of urbanization in proximity is much lower than that found in The Netherlands and the United Kingdom. The Scandinavian countries, two of the three Baltic states (Lithuania and Latvia), and Portugal are in a completely different situation from those described above. In these cases, the FRD50 index values are above 1 km (highest in Latvia with more than 1.6 km), and the UD values are extremely low. Therefore, in these cases, spatial reconnection measures that work on wider territorial scales (inter-municipal or provincial) are required because the distances to be overcome by spatial reconditioning are such that they cannot be incorporated into individual or coordinated groups of projects.

## 5. Discussion

This research analyzed the urban development affecting the various Natura 2000 networks in the countries covered. It was found that 1 km from the network band has the highest UD value and this is, in any case, higher than the national value. Furthermore, the UVR index highlights that the greatest variations affect the area inside the network sites, with Poland developing an area that is almost as large as the whole of Warsaw. However, it should be remembered that, while the CLC data is the only homogeneous European-level database (for thematic accuracy and acquisition techniques), the 25 hectares of MMU means that all the smaller urban structures were not calculated. This issue has already been dealt with in various studies [13–16] and strongly depends on the urban development models used by various countries. In Italy, and other countries, where settlement forms are scattered and frequently consist of just a few buildings, this value is certainly underestimated [17–19]. In countries such as the UK, France, and Germany, where the urban geography is much more compact and planned, the UD value reflected by CLC is much

more realistic. It should also be remembered that the UD index value is formulated by considering the relative proximity band [11] for the whole Natura 2000 network. Analysis of the individual sites would potentially provide more detailed information by subdividing the various proximity bands into radial sectors. This method can be used to obtain a picture of the possible interference generated by the area's settlement geography. A further factor that plays an important part in determining lesser or greater anthropogenic pressure on a given country's network is its morphology. Large flat areas foster development, and network sites in areas such as these tend to form part of considerably urbanized areas. This is tangible in countries such as Belgium, Germany, The Netherlands, and the United Kingdom in addition to Poland and Hungary, the location of one of Europe's largest plains, the Pannonian Plain.

## 6. Conclusions

In Europe, ecology networks are the main tool used in biodiversity conservation. Ensuring correct functioning over time and limiting the direct and indirect interference caused by urban development for the maintenance of biotic flow are thus essential. This study pursued these objectives, providing a good level of analysis of the degree of anthropogenic pressure on the matrix outside the Natura 2000 network and assessing the extent to which it has been damaged in the European countries. The information produced in this work is extremely important for two reasons. Firstly, it summarizes the current environmental fragmentation conditions (FRR curve and FRD index) of the N2000 network for each European country. Secondly, it shows the anthropogenic pressure in the immediate vicinity of the network (UD index). Indeed, the used method allows the possible scale of environmental defragmentation interventions (urban scale or planning level) to be set and identifies the urbanization conditions at different distances from Nature 2000 sites. In contrast, the used proximity analysis does not consider the urban morphology and urban geography. Furthermore, the diachronic analysis of the recent growth of urbanized areas through the reading of the UVR index values showed that in most European countries, the greatest changes concerned the sites themselves and the first 1 km belt from them. It should not be forgotten that these sites represent reservoirs of biodiversity and are important providers of different ecosystem services on which, as widely confirmed worldwide, the quality of life on Earth depends. All the results of this study show that it is necessary to integrate the structure of the national ecological network into regulatory instruments that are capable of controlling and directing urban changes according to the real socio-economic needs and are in line with the ecological-functional importance of the soils in the structures of environmental continuity. As highlighted in this study, urban development creates challenges for the effective integration of nature protection measures. The absence of regional regulation acts or regional planning guidelines that place reasonable limits on urban growth could generate uncontrolled urban sprawl, causing irretrievable loss of biodiversity and environmental integrity. Obviously, further and more detailed study into the interference generated by the infrastructure network is now required together with surveys on the discontinuities present in this network. Certainly, an integrated spatial planning approach is a useful tool for ensuring better protection of Natura 2000 sites from further urban transformations.

**Author Contributions:** Conceptualization, F.Z. and C.M.; methodology, F.Z. and B.R.; validation, F.Z. and B.R.; formal analysis, C.M.; investigation, F.Z., C.M. and G.D.P.; original draft preparation, F.Z.; review and editing, C.M.; visualization, C.M. and F.Z.; supervision B.R. All authors have read and agreed to the published version of the manuscript.

**Funding:** This research received no external funding.

**Institutional Review Board Statement:** Not applicable.

**Informed Consent Statement:** Not applicable.

**Data Availability Statement:** The data presented in this study are available on request from the corresponding author.

**Conflicts of Interest:** The authors declare no conflict of interest.

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
