# Peer review of "Urban Growth and Habitat Connectivity: A Study on European Countries"

_sustainability, doi:10.3390/su142214689_

Round 1
Reviewer 1 Report
The method used is too simple. The methods used in the previous studies should be reviewed. The authors should clarify why they choose such a simple method and what are the limitations. Is there any better method?
Author Response
Thanks for your comment. Our method has been already tested in Italy country and in Italian regions (please, see references 10 and 11). Although simple, our method allows to set the possible scale of environmental defragmentation interventions (urban scale or planning level) and it allows to know the urbanization conditions at differents distances from Nature 2000 sites. In contrast, the our proximity analysis does not consider urban morphology and urban geography.
In the conclusion section, we have added these sentences.
Reviewer 2 Report
The paper is to determine the degree of landscape fragmentation caused by the urban areas towards the Natura 2000 network, with the aim of analysing how the current urban settlements geography could compromise their functionality. Some minor suggestion:
Introduction: should add description of European’s urban growth and habitat connectivity;
Results:Now, the results just show the fragmentation curves for Western European countries,I think some more word need to do, such as to categorize different style according to the fragmentation curves.
Discussion: The paper clearly emerges that the 1 km from the network band is the one with the highest UD value.suggest to identify of the threshold.
Other:The text in Fig should to be larger size.
Author Response
The paper is to determine the degree of landscape fragmentation caused by the urban areas towards the Natura 2000 network, with the aim of analysing how the current urban settlements geography could compromise their functionality. Some minor suggestion:
Introduction: should add description of European’s urban growth and habitat connectivity;
Reply: Thanks for your comment. We have added this sentence on the Introduction section:
Large parts of Europe have become fragmented because of the expansion of urban and transport infrastructure with important effects on the habitat connectivity. As under-lined by European Environment Agency (https://www.eea.europa.eu/ims/landscape-fragmentation-pressure-in-europe), every km² in the 27 EU Member States (EU-27+UK) comprises around 1.4 habitats and 27% of the land is considered highly fragmented.
Results: Now, the results just show the fragmentation curves for Western European countries,I think some more word need to do, such as to categorize different style according to the fragmentation curves.
Reply: Thanks for your comment. We have added a table that resumes the different styles based to National Countries fragmentation curves. These styles resume the different condition of Natura 2000 found for each analyzed country.
Discussion: The paper clearly emerges that the 1 km from the network band is the one with the highest UD value. Suggest to identify of the threshold.
Reply: Thanks for your comment. It is a very important theme. The threshold identification needs to know the urban morphology and urban geography on the different buffer distances. In this paper, we have studied only the quantitative aspect through Urbanization Density index.
Other: The text in Fig should to be larger size.
Reply: Thanks for your comment, we have increased the figures readability. Furthermore, we have updated the figures caption.
Figure 3 and 4: The x-axis shows the buffer distances analyzed (N2000 sites, 1 km, 2 km, 3 km, 4 km and 5 km) while the y-axis shows the relatives UD (on the left y-axis) and UVR (on the right y-axis) values.
Figure 6 and 7: The x- axis shows the buffer distances analyzed (N2000 sites, 1 km, 2 km, 3 km, 4 km and 5 km) while the y-axis shows the relatives FRR values.
Reviewer 3 Report
I have reviewed the document "Urban growth and habitat connectivity. A study on the European countries".
The manuscript presents good quality, the authors have made a design according to the objectives investigated.
Authors should only separate the discussion and conclusions into two paragraphs.
Good job.
Author Response
I have reviewed the document "Urban growth and habitat connectivity. A study on the European countries".
The manuscript presents good quality, the authors have made a design according to the objectives investigated.
Authors should only separate the discussion and conclusions into two paragraphs.
Good job.
Reply: Many thanks for your comment. We have separeted the two paragraphs.
Reviewer 4 Report
The paper is exploring urban growth and habitat connectivity in the case of European countries. The manuscript, in general, is well written and with the appropriate manuscript structure. The research shows similarities and differences between Western and Eastern European countries concerning urban development challenges. The topic fits the scope of the journal, and the case is relevant. The manuscript describes applied research which has practical value, and the results and methods used are clearly presented. Overall, I propose to accept the manuscript for publication in its present form. I propose replacing Figures 3, 4, 6 and 7 with the one showing fewer countries – the text seems too small (text is unreadable in a pdf version sent to a review). Maybe try to avoid citing the literature in the conclusion.
Author Response
The paper is exploring urban growth and habitat connectivity in the case of European countries. The manuscript, in general, is well written and with the appropriate manuscript structure. The research shows similarities and differences between Western and Eastern European countries concerning urban development challenges. The topic fits the scope of the journal, and the case is relevant. The manuscript describes applied research which has practical value, and the results and methods used are clearly presented. Overall, I propose to accept the manuscript for publication in its present form.
Reply: Thank you for your comment.
I propose replacing Figures 3, 4, 6 and 7 with the one showing fewer countries – the text seems too small (text is unreadable in a pdf version sent to a review). Maybe try to avoid citing the literature in the conclusion.
Reply: Thanks for your comments. We believe that it is necessary to show fragmentation curves of all European countries. For this reason, we have both enlarged the Figures and incresead the readability. We have updated the captions.
Figure 3 and 4: The x-axis shows the buffer distances analyzed (N2000 sites, 1 km, 2 km, 3 km, 4 km and 5 km) while the y-axis shows the relatives UD (on the left y-axis) and UVR (on the right y-axis) values.
Figure 6 and 7: The x- axis shows the buffer distances analyzed (N2000 sites, 1 km, 2 km, 3 km, 4 km and 5 km) while the y-axis shows the relatives FRR values.
We have also removed the citations in the conclusion paragraph.
